# Healthy Patients, Workforce and Environment: Coupling Climate Adaptation and Mitigation to Wellbeing in Healthcare

**DOI:** 10.3390/ijerph20227059

**Published:** 2023-11-13

**Authors:** Mark de Souza, Aunty Bilawara Lee, Stephen Cook

**Affiliations:** 1Division of Emergency Medicine, Royal Darwin and Palmerston Regional Hospitals, Sustainable Healthcare Committee (NT Health), Tiwi, NT 0810, Australia; 2Office of First Nations Leadership, Charles Darwin University, Brinkin, NT 0810, Australia; bilawara.lee@cdu.edu.au; 3Commonwealth Scientific and Industrial Research Organization, Darwin, NT 0828, Australia; stephen.cook@csiro.au

**Keywords:** climate change, sustainable healthcare, biophilic design, Indigenous cultural safety, climate adaptation

## Abstract

Climate change threatens the health of all Australians: without adaptation, many areas may become unlivable, in particular the tropical north. The Northern Territory (NT) health workforce is already under colliding operational pressures worsened by extreme weather events, regional staff shortages and infrastructure that is poorly adapted to climate change. The H3 Project (Healthy Patients, Workforce and Environment) explores nature-based interventions in the NT health sector aiming to strengthen the resilience and responsiveness of health infrastructure and workforce in our climate-altered future. The H3 Project engaged the health workforce, climate researchers and the wider community, in recognition that meaningful and timely climate action requires both organization-led and grassroots engagement. We recruited campus greening volunteers and sustainability champions to Royal Darwin Hospital (RDH) to develop strategies that enhance climate adaptation, build climate and health literacy, and incentivize active mobility. We implemented low-cost biophilic design within the constraints of legacy healthcare infrastructure, creating cool and restorative outdoor spaces to mitigate the impacts of heat on RDH campus users and adapt to projected warming. This case study demonstrated substantial cooling impacts and improved local biodiversity and hospital campus aesthetics. We collaborated with Indigenous healers and plant experts to harness the synergy between Aboriginal people’s traditional knowledge and connectedness to land and the modern concept of biophilic design, while seeking to improve hospital outcomes for Indigenous patients who are both disconnected from their homelands and disproportionately represented in NT hospitals.

## 1. Introduction

### 1.1. Hospital Operational Pressures, Built Environment and Local Climate

Australia faces widespread impacts from climate change. The frequency and intensity of heatwaves, flooding and other extreme weather events is increasing, posing acute and compounding risks to human health in the foreseeable future [1]. The Northern Territory (NT) capital Darwin’s tropical climate already exposes people to the risk of heat stress, which demands that extreme caution to heat exposure is exercised between the months of September and April [2] (see Appendix B). Maximum daytime temperatures over 35 °C currently occur 47 days per year on average. However, under “worst case” warming scenarios, the average number of “extremely hot” days by 2050 will more than treble to 179 days per year [3]. These conditions will place an additional burden on the NT’s health services, which are already struggling to manage operational demands [4].

Chronic exposure to heat, humidity and seasonally poor air quality has well-known impacts on population health, contributing to cardiovascular deaths, some cancers, respiratory illness and mental health problems [5,6]. These impacts are predicted to worsen with climate change [7]. Vulnerability to heat illness is increased with low-income status, comorbidity, extremes of age, poor acclimatization and exercising or performing manual labor outdoors [8,9,10]. 

The H3 Project was initiated by members of NT Health’s Sustainable Healthcare Committee in September 2021 in response to the exhaustion and low morale of front-line healthcare workers that stemmed from a prolonged COVID-19 response, staffing shortages, hospital overcrowding and workplace aggression. The period was marked by a rise in patient complaints and critical incidents which included numerous episodes of emergency department nursing staff being assaulted while providing care. Patient complaint themes included long wait times and a lack of privacy imposed by “double bunking” in cubicles [11].

At this time, it was observed that the Royal Darwin Hospital (RDH) precinct was failing to provide the hospital campus users with restorative and climate-resilient green spaces to recover from these challenges, particularly for Indigenous patients who preferred to be outdoors. The precinct was prone to the urban heat island effect due to a paucity of tree canopies and an abundance of paved terrain [12], resulting in surface temperatures that can exceed 51 °C [13].

The RDH campus is predominately occupied by buildings and carparks, which results in warmer surface temperatures than the natural landscapes within the adjacent Casuarina Coastal Reserve. This results in a greater risk of heat stress, particularly for the vulnerable population groups that commonly frequent hospital campuses [14]. Figure 1 demonstrates the dry seasonal heat profile of different vegetated areas around RDH campus, reflecting the temperature difference between the campus and the monsoonal rainforest regions located to its west and south. Monsoonal rainforests remain relatively cool throughout the dry season, while areas of open eucalypt woodland and grasslands are relatively warmer [15]. Urban landscaping adopts shade trees and irrigated green spaces to reduce surface temperatures, which improves thermal comfort and can reduce building energy demand [16]. 

The expansive car parking at RDH reflects the dominant culture of motor vehicle commuting to work, with 74% of workers commuting by car and more than 60% of commutes being 10 km or less [18]. In contrast, active transport is known to promote heat acclimatization [19], reduce transport emissions [20] and improve physical and psychological health [21,22]. Arguably, the conditions on the health campus appeared to be undermining its therapeutic mandate and were impacting the wellbeing of healthcare staff who were working under intense pressure. Conversely, addressing these issues could potentially synergize with many NT Government strategies that aspire to promote workplace wellness, health and safety [23,24,25,26]. 

The H3 Project’s launch coincided with significant national debate about lagging Australian efforts to mitigate the health sector’s own contribution to environmental pollution and its carbon footprint, which is estimated to be 7% of total national emissions [27,28]. Australian healthcare workers had also declared their “alarm and frustration” at the lack of health-organizational response to climate change, with one study showing four out of five healthcare workers believed their health organizations should be “leading the way in climate action” [29]. The H3 Project sought to address these issues by linking climate action with wellbeing, focusing on strategies that highlighted immediacy of action and fostering climate and health literacy, local engagement and a mutually nurturing relationship with nature. Additionally, it called for accelerated cross-sectoral engagement in climate change adaptation, mitigation and healthcare decarbonization efforts at the organizational level.

### 1.2. Indigenous Perspectives and Context

RDH is the largest of the NT’s five hospitals, providing direct services to the Top End Region, and a tertiary referral service for the remaining regions and the remote Kimberley region of Western Australia. 

First Nations people comprise 30% of the NT population yet constitute 70% of hospital inpatients [30]. Indigenous patients at RDH have a 7–11% rate of leaving hospital before treatment is completed compared to 1–2% for non-indigenous clients [31], a trend which is nationally ubiquitous and arguably contributes to high rates of Indigenous morbidity and mortality. The causes of this national disparity in “self-discharge” are complex, but importantly include patients reporting isolation, loneliness and a lack of cultural safety in the hospital setting [32,33].

Seventy-five percent of Indigenous Territorians live in remote and very remote communities and continue to engage in traditional healing practices including the use of bush medicines [34]. These practices, which are one part of Indigenous people’s spiritual and cultural connection with their traditional lands, are highly place-specific and are integral to their “connection to Country” [35]. Remote-dwelling Indigenous inpatients indicate a strong preference for sitting outside the hospital, reporting air-conditioning to be “too cold” and a need to be unconstrained by walls and the built environment. The absence of a culturally safe outdoor environment was flagged in a recent national hospital accreditation report [36]. 

An attempt in 2017 to develop an Indigenous healing garden to improve Indigenous cultural safety at RDH was unable to reach its potential due to underfunding, attrition of project champions and issues with site accessibility [37,38]. 

The aim of this paper is to outline specific efforts undertaken to create climate-resilient, culturally safe spaces for staff and patients at RDH and to examine how successful responses have been developed through a staff-led process.

## 2. Materials and Methods

This case-study focused on grassroots sustainable healthcare initiatives and volunteer-led campus greening at Royal Darwin Hospital in the Northern Territory of Australia. Volunteer participation was essential to mobilize the hospital workforce at a time of fiscal austerity, while alleviating climate anxiety in the workforce and community members by creating opportunities for local climate action.

Figure 2 outlines the H3 Project’s phased implementation of low-cost, biophilic landscaping within the constraints of legacy campus infrastructure, funding restrictions and seasonal weather conditions. The project included engaging the hospital workforce, community and cross-sectoral executives and a call to recruit NT Health’s first Sustainability Officer to accelerate health sectoral efforts in decarbonization and waste reduction.

Phase 1 launched volunteer working bees within 2 months, targeting some zones accessible to all campus users and others used predominantly by hospital staff. We launched a volunteer recruitment and literacy-building program, consisting of internal staff communications (targeted email correspondence, electronic bulletins and live online forums) and a social media presence to reach the precinct’s multi-agency staff and the wider community. We commenced design of a sustainable healthcare intranet site intended to communicate progress and build engagement in future sustainability initiatives. 

Phase 2 and 3 consolidated and expanded the vegetation planting zones over the subsequent twelve months, accompanied by further efforts to recruit volunteers and engage health executives. 

We co-hosted a “Chalk the Campus” heat mapping event with the Darwin Living Lab on 26 May 2022. Over twenty participants representing a diverse range of staff and visitors were led on a 1-h tour of the precinct’s main walking routes and were asked to identify heat-prone areas and share their perception of how heat affects their activities and their health while using these areas. Participants marked these “hot spots” on maps and on the pavement, then outlined the cooling interventions that they believed should be applied in these locations. 

We launched a synergistic active mobility campaign, scheduled for the cooler dry season to maximize participation. Multi-agency staff and medical students were targeted, highlighting the environmental and wellness-promoting benefits of cycling to work, and of spending time and volunteering in the project’s nascent green spaces. 

### 2.1. Data Collection

Surface temperatures have been widely accepted as an indicator of urban heat across different land surfaces [39]. However, outdoor thermal comfort and heat stress risks are influenced by ambient air temperature, radiant temperature, relative humidity and wind speed [40]. 

The relative cooling impacts of the greening interventions were assessed with land surface temperature measurements using a thermal imaging camera (FLIR T530, Teledyne Technologies, Thousand Oaks, CA, USA) that also recorded image geolocation and orientation. Microclimate monitoring was also undertaken using the Scarlet TWL-1S heat stress meter (Scarlet Tech, Taiwan). This device measured air temperature, relative humidity, radiant temperature and wind speed, and calculated wet-bulb globe temperature as an indicator of heat stress conditions. 

Prior to planting, we performed two sets of baseline surface temperature measurements (Figure 3), the first corresponding to the dry season (23 August 2021) and the second to the wet season (15 December 2021). The first measurements were taken along Nightingale Road, in the location of a staff pedestrian and cycling corridor, close to staff entrances to both public and private hospitals and one of the main entrances to the public mental health unit. The second measurements were taken in a circuit around the main campus which corresponded to pedestrian and cycling routes for staff, patients and visitors. Both data sets incorporated a range of land surface types, irrigated/unirrigated zones and shading conditions (see Appendix C). Heat stress measurements were taken during the wet season on 11 March 2023 to compare heat stress conditions above different land surfaces (grassed and bitumen). Measurements were taken at 1.1 m height, which represents the average center of gravity for adults [41]. 

Follow-up measurements were taken in Planting Zones 2 and 14 to determine the effect of planting in reducing land surface temperature and heat stress as the trees established. 

Biodiversity impacts were documented by conducting bird surveys on the hospital campus, including in the vicinity of a significant banyan tree that had been scheduled for removal. Volunteer bird enthusiasts used a Google Drive document to record data from five biodiversity surveys which they conducted between April and November 2022. Personal correspondence detailing native bird sightings in other locations on campus were also included. We also collated the number and variety of native plant species that were introduced to the campus during the project and the plant death rate eighteen months after the start of planting. 

Qualitative data on perceived wellbeing and engagement impacts were informally gathered by collating personal correspondence and social media posts from volunteers, health workers, patients and the community. Formal data collection was outside the project’s resource capabilities.

### 2.2. Rationale for Site Selection

Vegetation planting zones are shown in Figure 4. With the exception of five Planting Zones (3, 6, 7 and 12 to 14), which contained occasional shrubs or trees, all our intervention sites were bare prior to planting.

Table 1 summarizes the rationale for inclusion for each planting zone during the first 18 months. Factors that favored early prioritization included proximity to gathering places, hospital entrances, end-of-trip facilities, footpaths and cycling routes, particularly when coupled with poor aesthetics and high surface temperatures at the location. We targeted sites close to entrances and walkways to maximize the frequency of biophilic encounters for busy staff and patients. Consideration was given to sites where soil erosion and weed infestation were problematic and those at which new restorative gathering places could be established through planting. The location of underground services (water, electricity, sewerage and storm water drains) and the availability of funding or donated items to allow work to proceed were further considerations. 

### 2.3. Species Selection

We postulated that replanting predominantly regional native plants would have numerous ecological, therapeutic and spiritual benefits. “Top End” plantings would pay homage to the original flora that had been cleared during colonization, be most likely to thrive in the soil and microclimates of the site and would best encourage the return of native fauna through the replanting of suitable food plants and habitat. Examples include the Darwin woollybutt (*Eucalyptus miniata*) and many species of grevillea which produce nectar-rich flowers and hollows to support native bees which in turn are key plant pollinators and food species for other native animals [42]. Host plants for the vulnerable Atlas moth were also selected, including the Atlas croton (*Croton habrophyllus*) and the Atlas moth plant (*Pittosporum moluccanum*) [43]. A variety of native grasses were also included to provide habitat and food for local species of finch and other birds.

We consulted with Senior Larrakia Elders who endorsed the project’s merit in seeking to enhance Indigenous health outcomes and wellbeing. They provided ethnobotanical expertise including a list of thirty-eight native Australian plants which were favored as bush foods and those associated with an ongoing practice of Larrakia ceremony and traditional healing. Of note was their recommendation to include ironwood (*Erythrophleum chlorostachys*), or “Mindilima” in the Larrakia language, owing to its foundational role in smoking ceremonies. When burned during the iconic Larrakia “Welcome to Country” ceremony, ironwood leaves emit a fragrant smoke that is believed to impart cleansing, protective and healing effects on participants as they walk through or waft the smoke about their bodies. We argue that offering our Indigenous inpatients—70% of all inpatients—greater opportunities to participate in smoking ceremonies and other complementary traditional healing practices may enhance the cultural safety of healthcare at RDH.

Over the subsequent 14 months, project leads sought advice on species selection from local experts in native plants and birds, supplemented with reference texts on local ethnobotany [44,45,46,47]. Further species were added upon recommendation by Indigenous patients and their family members. 

## 3. Results

### 3.1. Cooling Impacts 

Thermal imaging data taken periodically over the project showed a reduction in land surface temperature of up to 29.0 °C compared to the surrounding paved surfaces (Figure 5 and Figure 6). Figure 5 demonstrates the relative cooling benefits of greening intervention in Planting Zone 2. These images demonstrate the transformative effect of greening over an eighteen-month period in a wet/dry tropical climate, which reflects the correct selection of tree species for the sites and the use of dry season irrigation to maintain plant growth.

The results demonstrated that outdoor thermal comfort, as measured using wet-bulb globe temperature, did not vary significantly at locations with full sun exposure (Figure 7). This reflects that there were negligible differences in ambient temperature, relative humidity, radiant temperature and wind speed when undertaking paired observations for sun-exposed areas in comparison with green spaces and bitumen-sealed carparks. However, when comparing areas shaded by tree canopy and unshaded locations there was a substantial difference in the surface temperature. The shading of paths and other surfaces by trees improves outdoor thermal comfort and mitigates urban heat island effect by reducing absorption and re-emittance of solar radiation.

### 3.2. Biodiversity Impacts

#### 3.2.1. Flora

More than 950 substantive plantings occurred in the first eighteen months of the project. One hundred and forty-three Australian native species were planted, of which more than 90% were native to the Northern Territory (Appendix D). We estimate an additional 120–150 ground cover plants and shrubs have been propagated on-site from seeds and parent plant cuttings due to the success and cost savings associated with these methods for certain species. Sixteen exotic species were planted in low-light conditions of Planting Zone 15 which were less suited to native species. We observed a plant failure rate (plant death or removal due to failure to thrive) in the order of 3.1% (30 plants). We initiated a campaign to conserve an established banyan (*Ficus virens*) which had been scheduled for removal northwest of Planting Zone 7, at the site of the hospital’s new mental health development (refer to Figure 4).

#### 3.2.2. Fauna

Sightings of eleven Australian native bird species were documented at the site of this banyan tree during five separate surveys [48] that were conducted between April and November 2022 (Appendix E). In addition, there were sightings of multiple other native bird species including the endangered Gouldian finch (*Erythrura gouldiae*) in Planting Zone 6 and on a cycle path in close proximity to Planting Zone 8 [49,50]. 

### 3.3. Stakeholder Engagement

At the time of writing, more than 260 individuals have joined the RDH Campus Greening Volunteers social media group [51]. Members include multi-agency health precinct staff (healthcare, education and research organizations), medical students and diverse members of the community. Posts were also issued on a health agency-wide social media group with more than 4500 members [52] and to several local and national plant expert groups [53,54,55] to communicate progress, promote engagement, seek expert advice and attract volunteers. Each post to these sites was followed by new volunteer sign-ups, significant positive feedback and advice from a range of native flora and fauna experts that included species identification, pest management and networking opportunities (see Appendix F). Several posts attracted more than 100 likes and positive comments within 48 h of posting. Informal feedback was also received via email correspondence and direct verbal feedback to the project leaders. 

Feedback universally expressed gratitude, encouragement and congratulations for improving the physical conditions and aesthetics of the campus [56]. Many staff members shared positive reactions to the project, including one who reported parking their car “in the furthest [carpark] to take the longest route through the gardens” [57]. Social media posts on the biodiversity impacts of the project have attracted particular approval in the posts.

Twenty-four “working bees” (communal working events) were held over the first fourteen months of operation and were attended by between five and thirty volunteers, who were aged between three and sixty-five years old. 

Since the start of the project, many individuals, local businesses and non-government organizations have offered their support. These have included donations of seeds, mulch and plants, funding for planting activities in Zones 12 and 13 and notification of the availability of community grants. Participants and contributors emerged following promotion of the project though internal hospital channels, mainstream media reports and presentations at national climate health forums. The appearance of executive staff members and ministers at working bees is also likely to have stimulated interest. 

The “Chalk the Campus” event played a key role in building engagement in the workforce, executives of the precinct’s multiple agencies and in the community [58]. A diverse range of campus users shared their experiences of existing “hot spots” and suggested user-oriented solutions for climate adaptation and mitigation (see Table 2). Participants offered solutions that included providing shade infrastructure and planting at sites where pedestrian and cycling activity was greatest. 

Our campaign to conserve the established banyan tree elevated discourse regarding the interconnected priorities of cultural safety, wellness-promotion through biophilic design and biodiversity preservation amongst stakeholders for health infrastructure design. There is ongoing discourse with the NT Department of Infrastructure to improve active mobility infrastructure on the health campus [59,60].

### 3.4. Workforce Impacts

The H3 Project spearheaded the successful campaign to recruit the Northern Territory’s first Director of Sustainability Action. By the time of writing, the newly appointed executive had commenced mapping grassroots initiatives, recruiting additional sustainability officers and drafting a roadmap for environmental sustainability within NT Health. 

Hospital operations managers have modified their teams’ work plans to support the project. A greenhouse project has also been established at RDH to facilitate propagation activities on-site.

### 3.5. Wellbeing Impacts

Areas frequented by staff, patients and visitors were successfully planted, including active mobility thoroughfares, gathering places and highly industrialized zones where non-clinical support staff perform manual work outdoors. The outdoor seating infrastructure that was installed in the shade of Planting Zones 3 and 15 continues to generate positive feedback and is being utilized regularly by staff. 

Staff feedback obtained during the “Chalk the Campus“ heat-mapping workshop allowed “hot-spots” to be specifically identified and targeted. Participants represented the major healthcare, research and educational institutions operating on campus. 

Staff and visitors have enjoyed multiple harvests of edible rosella flowers (*Hibiscus sabdariffa*) and passion fruit (*Passiflora edulis*) which have been interplanted in several zones. Crop gathering encourages hands-on interaction with the green spaces while the self-sowing rosella species will provide sustainable harvests in the future.

### 3.6. Perspectives of Indigenous Patients and Visitors

We have successfully planted seventeen of the thirty-eight species and region-specific cultivars that were recommended by our Larrakia plant experts and Indigenous consumers. Red bush apple (*Syzygium suborbiculare*) was a strong favorite for the Larrakia, West Arnhem and Tiwi Islander peoples we interviewed. We included a pink cultivar of *S. suborbiculare* which is endemic to the Tiwi Islands and prized for its unique taste. Other popular requests included green plum (*Buchanania obovata*), billy goat plum (*Terminalia ferdinandiana*) and “an-binik” (*Allosyncarpia ternata)*. The distinctive boab (*Adansonia gregorii*) was planted to acknowledge Indigenous patients from remote areas of the Northern Territory and Western Australia who also receive care at RDH. Importantly, five specimens of ironwood (*Erythrophleum chlorostachys*) are now growing on campus whose leaves will be harvested for on-site Larrakia smoking ceremonies. 

### 3.7. Active Mobility Campaign Outcomes 

The campus’s main bike shed underwent significant landscaping and a tool station was installed to support minor bicycle repairs. A four-week ride-to-work campaign was held during the dry-season to maximize participation. Key elements included regional communications on active mobility, a staff E-bike raffle and a ride to work event at both the Royal Darwin and Palmerston Regional Hospitals. Proceeds from the raffle were gifted to the campus greening fund. There is an ongoing advocacy campaign to upgrade cycling and pedestrian infrastructure on campus including drafting a network of walking routes for display on hard signage and the hospital intranet (see Figure 8).

## 4. Discussion

### 4.1. Benefits of Green Spaces in Health Campuses

This study provides qualitative evidence indicating that increasing the tree canopies across the RDH precinct may provide substantial cooling benefits to campus users. This is consistent with previous studies [61] and highlights the importance of using tree canopies to shade surfaces and people from direct solar radiation.

Our heat mapping data indicates that the RDH campus exposes staff, patients and visitors to hot conditions as they traverse routes from outlying buildings and carparks to hospital entrances. This experience is shared by staff and clients of numerous co-located organizations including a private hospital, mental health clinic, childcare facility, university and a health research institute. In contrast, the grounds of the newer Palmerston Regional Hospital (circa 2018) featured comprehensive shade infrastructure and green spaces via a network of external and courtyard gardens. 

Tree canopies reflect and absorb solar radiation, which reduces the absorption and re-emittance of heat from surfaces such as roads and footpaths. The inclusion of well-irrigated green spaces can also provide cooling benefit through evapotranspiration [62]. The cooling benefit from evapotranspiration can vary significantly depending upon tree morphology [63,64] and the irrigation regime [65]. Mature, deep-rooted trees can often access moisture from deeper in the soil profile compared to shallow-rooted groundcovers, thereby maintaining evapotranspiration and cooling their surroundings even during extended heat waves [66]. Our project has selected a breadth of native species which, upon maturation, will provide a range of upper canopies, understory and ground coverage.

In addition to cooling healthcare precincts, the broader restorative benefits of green spaces have become well established in recent decades [67,68]. One systematic review of the literature found negative correlations between exposure to green space and mortality, heart rate and violence [69]. People living in urban areas are increasingly becoming disconnected from the natural environment, which may adversely affect their health and wellbeing [70]. The quality of the green space is also important in maximizing wellbeing, with more natural and biodiverse spaces providing more benefits [71]. Providing high-quality green spaces throughout the campus provided cool spaces for staff and patients to relax, build social connections and connect with nature [72]. High-quality green spaces improve outdoor thermal comfort and improve wellbeing through increasing the frequency of engagement in active transport modes such as cycling and walking [73].

Responding to warming predictions by simply providing access to more airconditioned refuges is inconsistent with emissions reduction, unless it is powered with renewable energy. It is also impractical due to prohibitive capital and operating costs and the potential return of social distancing restrictions in the future [74].

With the scientific evidence for global warming accumulating [75], the discourse has shifted to risk management and climate adaptation of assets and service delivery. There is therefore mounting pressure for both government and non-government agencies to be held accountable for making adequate provisions to counter the health impacts of climate change [76]. 

The H3 Project identified the omission of the RDH campus from two major climate adaptation strategies that sought to improve resilience to the impacts of climate change in the region [77,78]. Despite the harsh local weather conditions, the grounds of the NT’s only tertiary healthcare facility lacked a clear strategy for increasing tree canopy cover and other greening to mitigate heat stress risks and adapt to a warming climate. We engaged and secured research support from the Darwin Living Lab, a project developed under the auspices of the Darwin City Deal [79] to progress sustainability and liveability in Darwin.

Investment in healthcare green spaces can offer significant protection from the current heat stresses that prevail in the Australian tropics and is paramount given the extreme warming predictions for the region [80]. Climate change poses a significant risk to the stability of the NT’s health workforce, which is reliant on twice-yearly recruitment from southern Australian and international jurisdictions. The January–February staff intake is responsible for the bulk of new recruits and coincides with the wet season, which is notorious for high temperatures and relative humidity. New healthcare recruits to the NT who arrive without adequate acclimatization are more vulnerable to the spectrum of heat-related illness, impaired productivity and sick leave [81]. There is also evidence that climate change is exacerbating healthcare workforce shortages in underserved rural and remote areas of the NT [82]. 

### 4.2. Biophilic Design and Human Health

First coined by Fromm in the 1960’s, biophilia is the innate human instinct to connect with nature and other living beings [83]. The concept of biophilic design is an architectural approach to designing the built environment that enhances human functioning through nurturing connectivity with nature [84]. There is now extensive implementation of biophilic design in educational institutions [85] and office-based businesses [86] to maximise productivity and counter the negative impacts of unsustainable urbanization and the rising incidence of stress-related conditions and their ensuing health expenditure. 

Research in the past decade compels the health sector to implement biophilic design from staff, patient and organizational perspectives. Observed patient benefits include reduction in blood pressure, post-surgical recovery time and the need for pain medication, while improving mood and satisfaction with health services [87]. Emerging evidence shows the potential benefits of “nature prescriptions”, such as “forest bathing”, for ameliorating cardiometabolic disease [88]. Elderly patients, including those with cognitive impairment, may receive a substantial therapeutic impact from engaging in green spaces through mechanisms that include reinforcing a sense of identity and empowerment, enabling meaningful engagement and positive risk-taking [89]. The benefits of engaging in therapeutic horticulture during hospital recovery may accrue through encouraging physical and mental exercise and facilitating a nurturing interaction with nature [90].

Importantly, biophilic encounters in the healthcare setting are most restorative when they assist individuals to contemplate the existence of a reciprocal, nurturing relationship with nature [91]. These encounters should promote awareness of nature’s essential role in restoring human health and wellbeing, while human beings can and should embrace lifestyle changes that show mutual respect for nature’s restoration. Additional evidence suggests that biophilic encounters that emphasize the human–nature connection can directly promote feelings of happiness [92]. 

The native landscapes developed by this project satisfy the conditions that may trigger this contemplation of nature. These include offering a spectrum of visual and non-visual stimuli, a variety of far and near visual perspectives and both ordered and complex stimuli [93]. Notable examples are the natural soundscapes of birdcalls and moving water and the variety of fragrances and leaf forms offered by the diversity of planted species. Evidence indicates that spending even five to twenty minutes in such spaces may result in a restorative effect on physiological stress indicators [94] and anxiety scores [95], while repeated visits are likely to have a cumulative effect [96]. 

### 4.3. Organizational Benefits of Green Spaces

Working in healthcare is inherently stressful, with one pre-pandemic study revealing 40% of staff had felt unwell as a result of work-related stress within the previous 12 months [97]. The same study showed around 90% of staff wanted to spend more time in green spaces than they currently did, with a majority citing benefits such as feeling relaxed and calm, more productive on returning to their duties and feeling positive impacts on their physical and mental wellbeing. Australian efforts to embed workforce wellness programs have quickened in recognition of the converging benefits of promoting health in an ageing workforce while optimizing productivity and reducing population healthcare costs [98]. Healthcare staff who regularly spent time or undertake activities in green spaces at work rate their wellbeing as significantly higher than those who do not [99]. Activities include taking food breaks or a short walk, attending meetings and organized outdoor activities and engaging in spiritual activities. Other studies have shown both staff and patients seek out green spaces “for fresh air, to talk with family and to express emotion” [100]. The same researchers found biophilic design influenced recruitment and retention of health workers, with around 50% of respondents indicating that access to attractive green spaces was important in considering new employment. This has implications for the stability of the NT health workforce, which relies heavily on regular intakes of interjurisdictional recruits and an increasing dependence on locum staff.

Healthcare staff also desire access to green spaces that are reserved solely for their use, where they can be assured of being “off the clock” and out of the eye of patients and carers [101]. Locations such as rooftops and courtyards are ideal, while also providing added visual perspective. Some staff identify outdoor green spaces as the ideal location to discuss lifestyle changes, provide counselling and give bad news to their patients. Access to green spaces in which natural phenomena can be observed (such as the changing of the seasons) can assist individuals to engage in reflection and come to terms with challenging personal circumstances [102], a feature which would appear indispensable for anyone receiving or providing healthcare. 

Accessibility for sight- and mobility-impaired users should also be considered, as well as infrastructure that promotes security and safety. Our project sought to overcome local barriers to accessing healthcare green spaces. Champion- and consumer-led assessment of hot spots, focusing on active mobility thoroughfares and consideration of typical staff, patient and visitor journeys at RDH resulted in site selection that maximized opportunities for biophilic encounters. 

### 4.4. Benefits of Participation in Land Care Activities and Biodiversity Surveys

On a basic level, our greening interventions improved the amenity of a health campus that was impacted by weeds, soil erosion and lack of irrigation. Importantly, healthcare staff and the community were encouraged to participate in local land care activities whose known social, cultural and learning benefits are summarized in Table 3 [103]. These collective and locally-executed activities enhance participants’ relatedness to nature while helping to alleviate eco-anxiety [104,105,106] in a region known for its unfolding fauna extinction crisis [107] and encourage vigorous debate on the environmental impacts of land-clearing practices and new fossil fuel projects [108,109]. Furthermore, participating in land care activities combats environmental generational amnesia, wherein environmental degradation becomes normalized with each successive generation [110]. 

The project’s early biodiversity impacts are worth highlighting. Enthusiastic social media posts, participation in bird surveys and reports of Gouldian finch sightings point to strong community and health workforce interest in implementing conservation activities on the RDH health campus. The results of our biodiversity survey and the Indigenous cultural significance of *Ficus virens* [111] were key arguments in our campaign to conserve an established specimen and its attendant fauna. This conservation campaign elevated organizational discourse on the wellness benefits of biophilic design in NT healthcare assets and the cross-promotion of local biodiversity.

### 4.5. Promotion of Indigenous Cultural Values

The H3 Project continues to consult with Larrakia Elders who represent the traditional custodians of the land on which RDH stands. We noted that older campus plantings consisted mostly of exotic ornamental species which provided no link to Indigenous culture and were unsuccessful due to improper species selection, pest incursion and unmet demands for water and maintenance. 

This partnership identified native plant species that are favored by the Larrakia people, such as bush foods, and for traditional healing and ceremony. We showcased ironwood due to its foundational role in smoking ceremonies. Offering our Indigenous inpatients—70% of all inpatients—greater opportunities to participate in smoking ceremonies and other complementary traditional healing practices will enhance the delivery of culturally safe healthcare. 

The RDH also treats patients from many distinct Indigenous groups across the NT as well as the Kimberley region of Western Australia. A concerted effort was made to plant native species from these areas wherever permitted by the climate and soil conditions of the campus. Informal feedback from Indigenous patients and their escorts supported the planting of familiar and regionally specific species to help provide a connection to Country during prolonged separations. 

Our project’s scope includes the curation of online resources and physical signage to highlight the Larrakia names and traditional uses of significant species. This component aims to promote awareness and respect for Indigenous culture in staff, patients and visitors. In turn, it makes the health campus more welcoming to Indigenous people.

## 5. Limitations

After fourteen months of operation, only preliminary heat mapping and qualitative wellbeing impact data have been obtained. Several years’ vegetation growth and dedicated research funding is required for the cooling, aesthetic and wellbeing impacts to be fully evaluated. Similarly, the biodiversity impacts of the project to date are qualitatively assessed and likely to be underestimated. While the H3 Project achieved a significant visual transformation of the RDH campus, any further expansions and nurturing of existing plants to maturity requires dedicated funding and support from an employed workforce.

## 6. Future Directions 

We plan to install a weather station on RDH campus to facilitate ongoing data collection for heat stress measurements, which will provide reference points for localized climatic conditions. We intend to install additional seating, drinking water outlets and campaign for the installation of additional shade infrastructure in areas regarded as “hot spots” by participants in the Chalk the Campus event. We are exploring mobile application-based interactivity for campus users to foster relatedness to nature.

Further plantings are being planned in the Indigenous accommodation precinct of RDH to improve thermal comfort and offer a “connection to Country” during prolonged separations. We plan to install ethnobotanical signage and establish a series of walking paths that will be promoted to staff and patients as “nature prescriptions”. We are looking for opportunities to collaborate with Larrakia ethnobotanists to conduct harvesting activities and eco-cultural tours. As plants mature, we intend to facilitate regular practice of on-site Indigenous ceremony and traditional healing. It will be vital to seek ongoing feedback on the wellbeing impacts of these curated landscapes on First Nations people and quantify any effect on the rates of leaving hospital before treatment is complete. 

We intend to facilitate engagement of healthcare workers with a volunteer land care group that operates in the coastal reserve adjacent to the RDH campus. This may encourage recruitment to local land care efforts while offering health workers opportunities to combat eco-anxiety and environmental generational amnesia.

Quantitatively assessing the full health, environmental and economic benefits of this case-study would assist health services to further understand and rationalize investment in similar projects in the NT and other jurisdictions.

## 7. Conclusions

Health campuses have the potential to alleviate the hardship of delivering and receiving care in the face of colliding operational and climatic pressures. 

Biophilic design offers low-cost adaptation of legacy health infrastructure to climate change that promotes biodiversity, wellbeing and a relationship of reciprocal nurturing between campus users and the environment.

This case study set in the harsh climatic environment of the Northern Territory provides a platform for improving cultural safety and hospital health outcomes for First Nations Australians, while promoting Indigenous knowledge and leadership in climate adaptation and mitigation efforts in the healthcare system.

## Figures and Tables

**Figure 1 ijerph-20-07059-f001:**
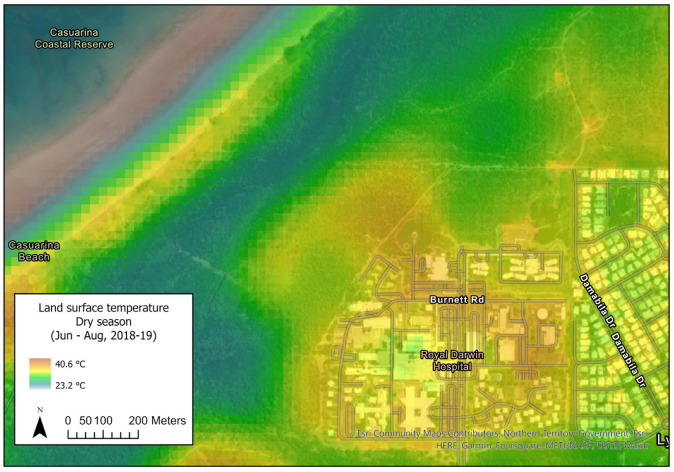
Land surface temperature (LST) at RDH campus. Source: Meyers et al. [17]. Mapping land surface temperatures and heat–health vulnerability in Darwin. Commonwealth Scientific and Industrial Research Organization (CSIRO), Australia [17].

**Figure 2 ijerph-20-07059-f002:**
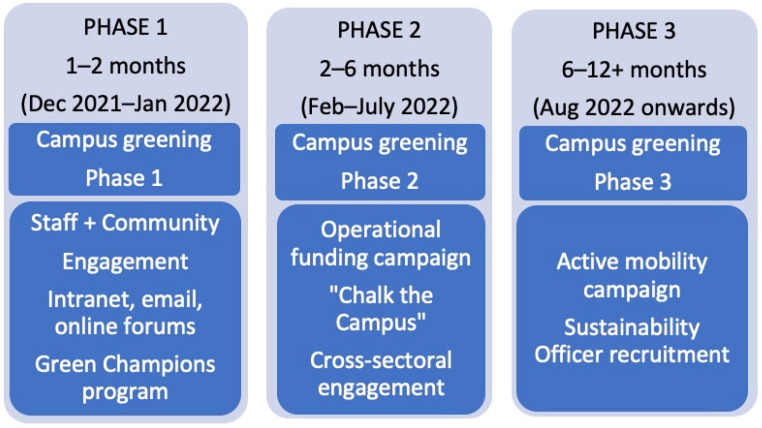
Implementation phases and key activities for the H3 Project.

**Figure 3 ijerph-20-07059-f003:**
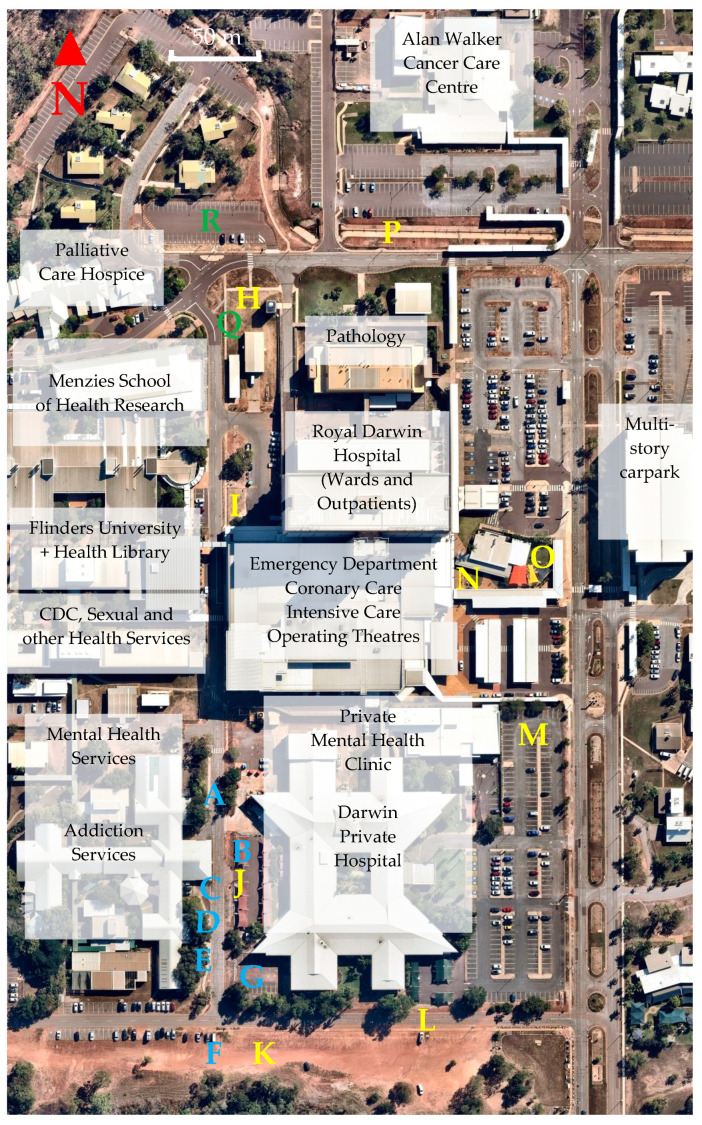
Baseline thermal imaging sites A–G (shown in blue) taken at RDH campus at 3:34 p.m. on 23 August 2021 (temperature 31 °C, relative humidity 56%) and subsequently at sites H–P (shown in yellow) between 3:00 p.m. and 4:14 p.m. on 15 December 2021 (temperature 35 °C, relative humidity 57%). Heat stress measurements (shown in green) were taken at site Q with exposed grass surface and site R with exposed bitumen surface between 3:05 p.m. and 4:05 p.m. on 11 March 2023 (temperature 32 °C, relative humidity 58%).

**Figure 4 ijerph-20-07059-f004:**
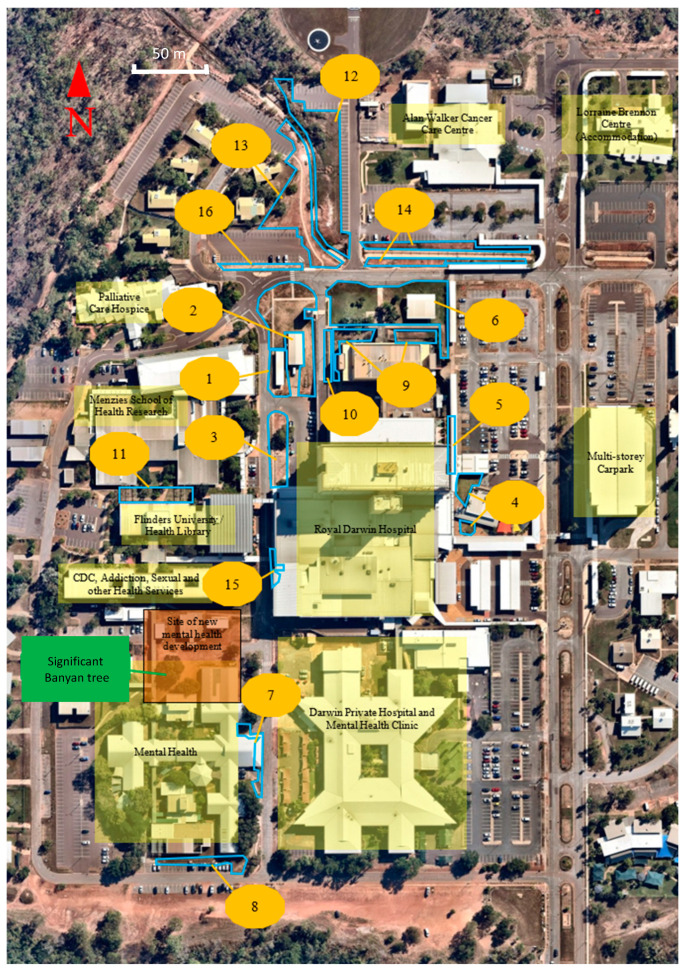
Planting Zones (labelled in yellow, outlined in blue) on RDH campus. 1. Nightingale bike shed; 2. Allied Health shed; 3. Engineering dock; 4. Main entrance lawns; 5. Carpark 1 walkway; 6. “Smoker’s pavilion”; 7. Cowdy Ward; 8. Mental Health Services (MHS) south entrance; 9. Pathology building lawn; 10. Pathology building green wall; 11. Health Library trellis; 12. Alan Walker Cancer Care Centre (AWCCC) culvert east; 13. AWCCC culvert west; 14. Burnett Road culvert; 15. Emergency Department (ED) rear entrance; 16. Carpark 5A footpath.

**Figure 5 ijerph-20-07059-f005:**
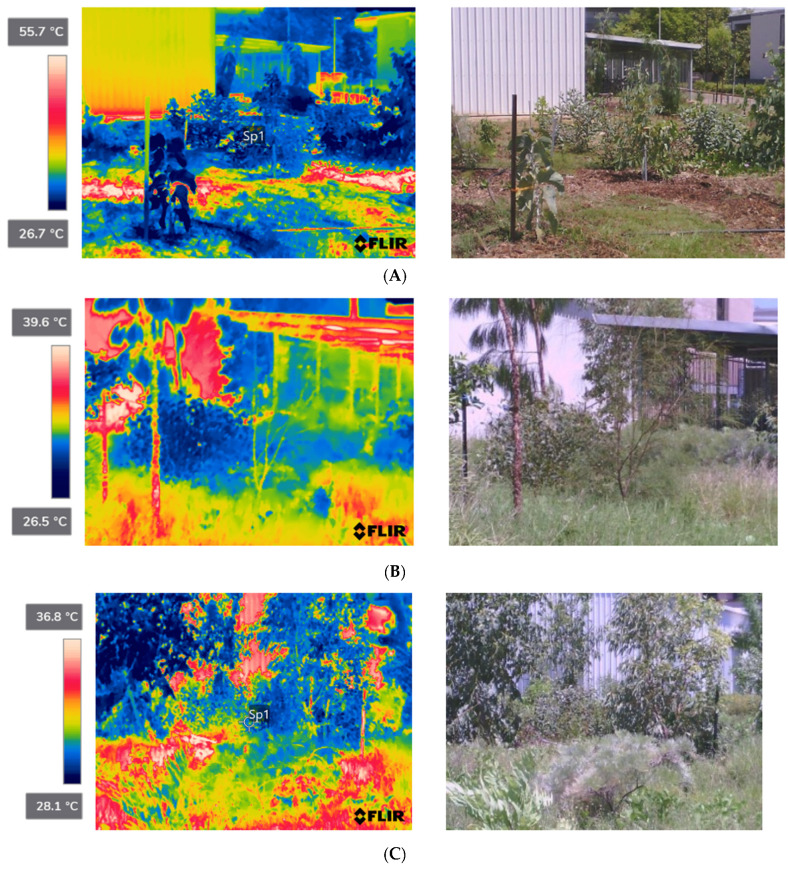
Relative changes in surface temperature in the Allied Health shed (Planting Zone 2) following greening interventions: (**A**) Dry season (16 September 2022 1:45 p.m.)—ambient temp. 33 °C, relative humidity 58%; (**B**) Wet season (11 March 2023 2:04 p.m.)—ambient temp. 32 °C, relative humidity 55%; (**C**) Wet season (11 Mar 2023 2:10 p.m.)—ambient temp 32 °C, relative humidity 55%.

**Figure 6 ijerph-20-07059-f006:**
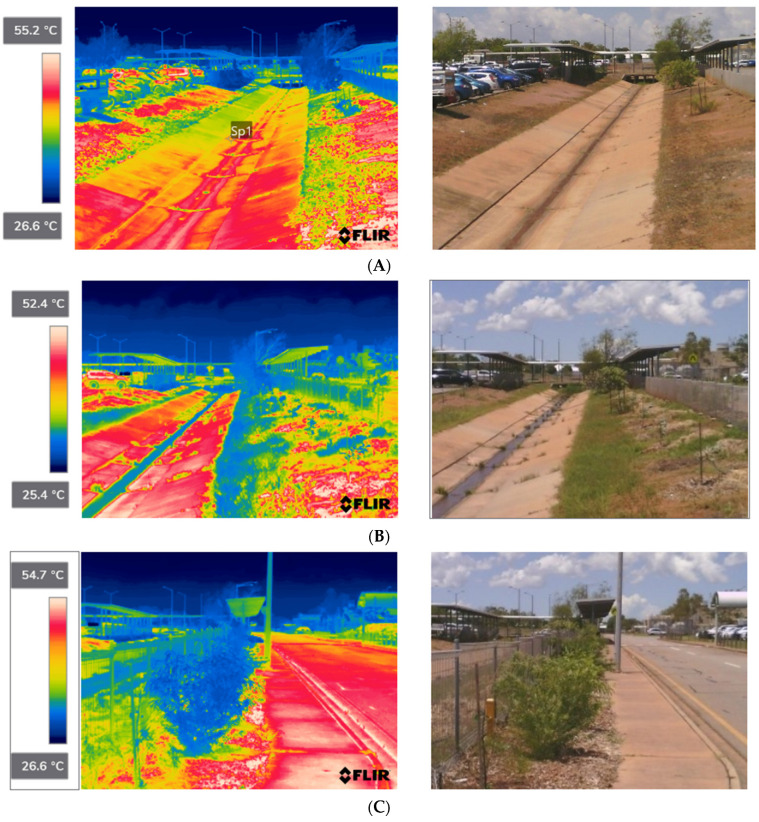
Relative changes in surface temperature in the Burnett Road Culvert and adjacent footpath (Planting Zone 14) following greening interventions: (**A**) Dry season (16 September 2022 1:38 p.m.)—ambient temp. 33 °C, relative humidity 58%; (**B**) wet season (11 March 2023 1:54 p.m.)—ambient temp. 32 °C, relative humidity 55%; (**C**) Wet season (11 March 2023 2:07 p.m.)—ambient temp. 32 °C, relative humidity 55%.

**Figure 7 ijerph-20-07059-f007:**
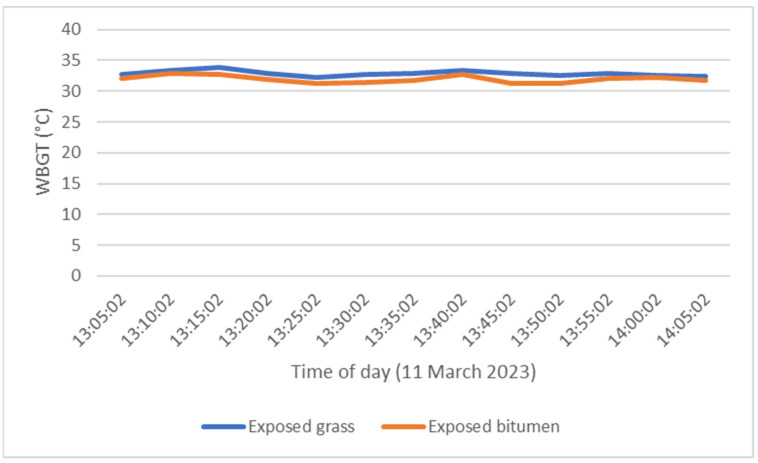
Comparison of wet-bulb globe temperature (WBGT), with full sun exposure on grassed surface (taken at site Q) and bitumen surface (site R) as shown in Figure 3.

**Figure 8 ijerph-20-07059-f008:**
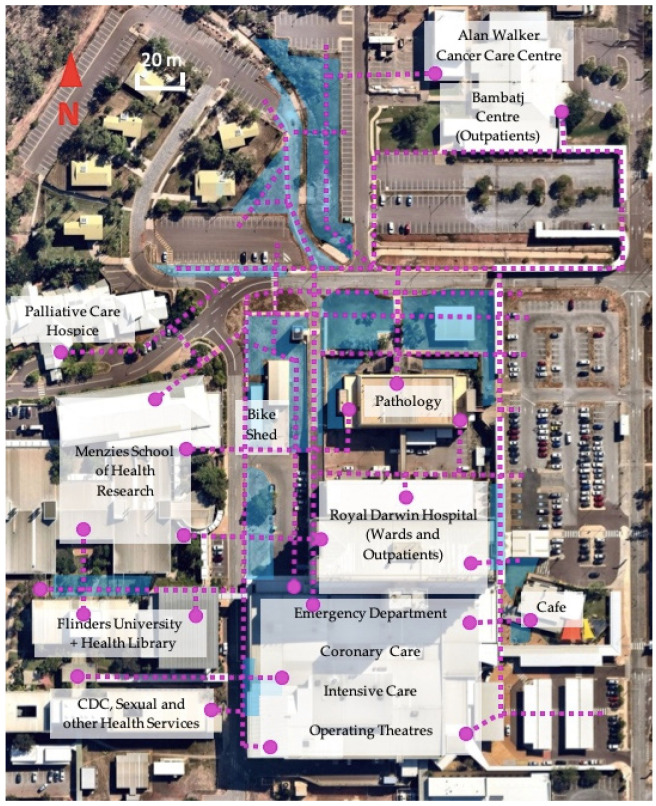
Location of the project’s developing network of walking routes (dotted pink line) linking buildings, gathering places and end-of-trip facilities. Walking routes are co-located along known active mobility pathways to maximize exposure to Planting Zones (blue shading). Note: Building entrances indicated by pink dot.

**Table 1 ijerph-20-07059-t001:** The H3 Project’s 16 Planting Zones, their phases of implementation and justifications for planting.

	Planting Zone															
	1	2	3	4	5	6	7	8	9	10	11	12	13	14	15	16
PHASE 1																
PHASE 2																
PHASE 3																
Improved aesthetics	+	+	+	+	+	+	+	+	+	+	+	+	+	+	+	+
Exposed to climate/UHI effect	+	+	+	+		+		+	+	+	+	+	+	+	+	+
Staff/patient request		+	+	+	+		+	+				+	+	+	+	+
Pedestrian/cycling route	+	+	+	+	+		+	+	+	+	+	+	+	+	+	+
Near building entrance		+	+	+	+		+	+	+	+	+		+	+	+	
Near end of trip facility	+	+	+	+	+		+	+			+	+	+	+	+	+
Multi-agency staff access to area	+	+	+	+	+	+		+	+		+	+		+	+	+
Outdoor work area	+		+	+						+		+	+	+		+
Staff gathering place			+	+		+			+	+	+				+	
Patient/visitorgathering place				+	+	+	+		+		+	+	+			
Potential newgathering place	+		+	+	+			+	+	+	+	+	+		+	
Potential site forblue space												+	+	+		
Prone to weeds (W)or soil erosion (S)		W	WS				WS		W	W		WS	WS	WS	W	WS
Replace dead orweak plants	+			+			+						+			
Requires new mainswater supply												+	+	+	+	+

KEY: Planting Zones: 1. Nightingale Road bicycle shed, 2. Allied Health Shed, 3. Engineering dock, 4. Main entrance lawns, 5. Carpark 1 walkway, 6. ‘Smoker’s pavilion”, 7. Cowdy Ward, 8. Mental Health Services south entrance, 9. Pathology building lawn, 10. Pathology building green wall, 11. Health Library trellis, 12. AWCCC culvert east, 13. AWCCC culvert west, 14. Burnett Road culvert, 15. Emergency Department rear entrance, 16. Carpark 5A footpath. AWCCC: Alan Walker Cancer Care Centre, UHI: Urban Heat Island, End of trip facility: carpark, bus stop, bicycle shed or bicycle rack. Shading denotes the phases of the H3 Project during which planting activity occurred in each Planting Zone. + denotes justifications for activity in each Planting Zone.

**Table 2 ijerph-20-07059-t002:** Participants, heat issues and solutions voiced during the Chalk the Campus event held in May 2023.

Professional and Community Group Representation	
Community climate action group member	Hospital operations managers
Community cycling advocate	Medical researchers
Community members (health service consumers)	Medical students
Emergency physician	Nephrologist
Environmental officer	Paramedics
Health librarians	Physiotherapists
Hospital administration officers	Security officers
Hospital facilities officers	University program manager
Heat issues raised	
Heat island effect from the large area of exposed carparks.	
Unshaded bare ground and footpaths radiate heat to pedestrians.	
Concern for impact of heat exposure on staff and patients with health and mobility issues.	
Lack of shaded active mobility routes discourages participation in cycling, walking, E-scooter modes of transport to work.	
Existing covered walkways only provide shade in the middle of the day (no eaves to block early and late sun)	
Lack of seating infrastructure in areas marked as cool refuges.	
Solutions offered by participants	
Need urgent planting interventions to build resilience to warming climate (trees take time to grow).	
Improved use of built infrastructure (covered walkways, include eaves to block early/late sun).	
Consider campus journeys of staff, patients and visitors when designing cooling interventions.	
Expand current shade infrastructure for walking and cycling.	
Integrate greening in campus master-planning.	

**Table 3 ijerph-20-07059-t003:** Benefits of land care activities. Reproduced with permission from Australian Dept of Agriculture, Fisheries and Forestry. Published by GHD Group in Multiple benefits of Land care and Natural Resource Management (Final Report for the Australian Landcare Council), 7 July 2013. Please see more information on Appendix A.

Benefits of Landcare	Sub-Categories
Learning, awareness and practice change	Awareness raising Practice change Multigenerational reach Improved knowledge Scales of change Continuous learning
Social: community health and wellbeing	Contact with natural environment Social networks Physical and mental health benefits
Social: political and social capital	Partnerships and networks Leadership and public participation Governance and self-regulation Localism and empowerment Increasing the recognition of women in rural communities Personal growth Filling the void Increasing awareness, skills and knowledge
Economic	Increased financial return Access to resources Training and management techniques
Cultural	Connection with Country
Resilience	Resilient people and resilient landscapes

## Data Availability

Data available on request from the corresponding author. Data are contained within the article and Appendix A.

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
