# Peer review of "Healthy Patients, Workforce and Environment: Coupling Climate Adaptation and Mitigation to Wellbeing in Healthcare"

_ijerph, 2023, doi:10.3390/ijerph20227059_

Round 1

Reviewer 1 Report (Previous Reviewer 3)

Comments and Suggestions for Authors

The manuscript was, in fact, revised according to my recommendations.

Author Response

Dear Reviewer

Thank you for your assistance and advice. 

It appears that we have now applied the suggested revisions.

Best regards

Dr Mark de Souza

Reviewer 2 Report (New Reviewer)

Comments and Suggestions for Authors

The manuscript submitted for review describes a project to introduce greenery in a large medical center. The Authors did not formulate a research goal and, in fact, did not conduct scientific research. They only announce that they will conduct surveys among users and contractors of the project in the described area in the future. The stages of work described in the methods are stages of project implementation, not stages of scientific research. The research performed on heat stress and biodiversity changes resulting from the project implementation is not sufficiently well documented, as well as the lack of a test field that can constitute a reference level. We have no information about the results of heat stress measurements in the project area before the changes began. Fig. 4 and Fig. 5 shows the well-known cooling effect of greenery, but the Authors did not provide specific details - what was the area and species of plants introduced in this place, what meteorological conditions existed on the day of the thermal imaging camera examination. Changes in biodiversity have not been documented by comparing the fauna and flora from the period before the start of the project with the current state. We also have to take the information about public reception and the positive impact of the project on people on their word; the Authors did not create a database documenting public opinions, such as e-mails or social media entries. Due to the above facts, I believe that the presented description of the implementation project does not have the characteristics of a scientific study. But - the information about the undertaken initiative and the observed positive implications is interesting and may be useful in planning the adaptation of other urbanized areas. Therefore, I believe that a piece of information together with a description of the project can be included in the journal as a report announcing future research.

Other reservations and specific questions worth considering in future research:

- results, lines 247-281: temperature reduction of the surface with greenery by 20oC - no description of meteorological conditions at the time of measurements, type and color of the paved surface, type, density and height of vegetation,

- impact on biodiversity, lines 282-303: the Authors refer to the participation of the local community in the selection of native species most suitable for a given place, and then describe the introduction of "foreign" species in connection with the use of the hospital by residents of other regions and the desire to create a friendly, close home environment. These are contradictory approaches, the latter approach is inconsistent with the principles of biodiversity conservation. There is no information on biodiversity in this area at the beginning of the project.

- The Authors write about the participation of participants aged 3 and over in the project (line 331) - the participation of children in research projects should take place with the consent of their parents/legal guardians. Do the Authors have the consent of the guardians for the children's participation in the above-mentioned project?

  - impact on well-being - the Authors do not provide any evidence for the described impact, when were the workshops held, how many people participated in them, where are the results of the interviews available?

- line 345 – how did project participants identify "hot spots" on campus? How many of these points were identified, were there differences in the indicated places, were all of them covered by greening?

The discussion in the scientific article should refer to the results of research, which, however, was not conducted in a scientific sense in this project. In my opinion, the discussion in this manuscript constitutes a kind of literature review of the issues addressed in the project. For example - lines 396-397: I did not find information about tree limb measurements in the results, I do not know on what basis the authors stated that "We have shown that increasing the tree canopies across the RDH precinct provides substantial cooling benefits to campus users." For example, in Fig. 4 and Fig. 5a, 5b there are no tree crowns.

In the discussion, the Authors refer to evapotranspiration, which they have not studied, to health status and mortality, which have also not been studied, to the recruitment of medical staff, and then broadly describe the impact of biophilic design on human health; they devote an entire paragraph to the relationship between green areas and COVID-19 - which they also did not investigate in their project.

Lines 544-545: the Authors write about feedback from staff and patients - but there is no information, where can you read it?

Line 557 – there is a comment about the medicinal garden project from 2017. This is the first information about this project, about which no further information is known.

Line 561 – a reference to two courtyard designs – the reader also knows nothing about them.

lines 582 – 583: the Authors write about posts in social media - no confirmation of their existence, no database with content, number, etc.

Author Response

Dear Reviewer

Thank you for these detailed comments.

The authors respectfully re-emphasise that this paper represents a case study, rather than scientific research per se. 

Overall, we wish to demonstrate that greening projects on health campuses using native species broadly improves wellbeing through promoting indigenous cultural safety, biodiversity, cooling and exposure to biophilic landscapes. 

We address your specific concerns as follows: 

While we did not calculate the surface area of the planting area or tree limb lengths, we included what we regard to be more important characteristics, notably the proximity of biophilic spaces to walking paths which link end of trip facilities and building entrances.  The aerial view of the campus and our planting zones (Figure 4, Line 229) provide the reader with this relationship, however we concede that a scale (in metres) would be helpful and have now included this.

Likewise, we were not intending to ascertain the differential cooling capacities between species of Australian native plants (though these are emerging areas of research in Australia), but that the mere presence of ground covers, mid story and upper tree canopies provided a significant cooling impact compared to the bare soil and paved surfaces that dominated the hospital's campus prior to our work.  We agree that the excellent technical variables you raise such as tree crown diameter and inter-species differences are worthy variables to interrogate in more robustly funded research projects in the future.

Regarding baseline land surface temperatures on campus we are pleased to now include two sets of baseline data that was collected by the third author in August and December 2021, taken during the Dry and Wet seasons. Amendments are as follows:

We have retained in the text a brief mention of baseline surface temperature at Lines 62-63 and added explanatory text regarding baseline data collection in Methods (Lines 174-186 ).

A new Figure 3 is included from Line 190 showing the location of baseline temperature readings and the  climactic conditions of the day.

New Appendix B added from line 656 which provides the relevant map locations for baseline recordings, meteorological data and thermal images. 

The contents of Figures 5 and 6 (Lines 279-301) have been revised to now demonstrate surface temperatures in Planting Zones 2 and 14, including individual min/max temperatures measured at each site.

A new Figure 7 at Line 302 is now included to show the comparison between bitumen and grassed area heat baselines using wet bulb globe temperature.

Detailed analysis of the heat absorption and remittance of surfaces of differing colour and material type is the subject of numerous other studies and was outside the scope of our project. 

Regarding baseline flora surveys, with the exceptions stated in Lines 216-218, our Planting Zones at baseline were essentially bare of any vegetation.

Thank you for your helpful suggestions regarding the "Chalk the Campus" heat mapping event.   We have now amended the method and future direction sections of our new submission (Lines 151-157, 606-609) and included a new Table 2 (from Line 361) to summarise the participant’s concerns and solutions voiced during this event.  

In response to the suggestion by Reviewer 3, we have included a new Figure 8 (Line 414) illustrating the network of active mobility corridors that are being developed through our planting.

We emphasise that the methodology of the heat mapping event was to permit participants to share their lived experience of hot areas on the campus and suggest user-oriented solutions. The event also provided individuals with an opportunity to influence local climate adaptation, participation in which is shown to contribute to relieving eco-anxiety (Line 556-559, Refs [104-106]).

All children aged less that 18years of age attended our communal working bees accompanied by their parents, indicating tacit parental consent.  As participation by adults and minors was part of our hospital volunteer project and not as research participants, written consent for working bee attendance was not required.

Regarding selection of species and biodiversity, we respectfully refer to our project aim to improve the cultural safety of our health campus for Indigenous patients (who are disconnected from their homelands when they travel to an urban hospital), by planting culturally significant native species.  Importantly, the dispersal of almost every Northern Territory native plant is not confined to one geographic area, rather there are pockets of their occurrence over large regions in the wild.  We provide the case of Acacia umbellata whose wide dispersal in the Northern Territory of Australia can be seen at https://apps.lucidcentral.org/wattle/text/entities/acacia_umbellata.htm.

We have been transparent regarding the rationale for the occasional planting of non NT natives and exotic species.

We do not set out to evaluate the rate of evaporo-transpiration, heat reflection or absorption that occurred in our project.  We briefly mention these to explain to our non-technical readers the means by which greenery is understood to deliver its cooling effect.

The authors have removed the reference to the two courtyards designs previously in Line 561.

We have also removed the paragraph on “COVID 19 and green spaces”.

Thank you for requesting to include staff and community feedback for our hospital project, the amount and positivity of which has been a major driver for our ongoing efforts.  We have now included the link to the social media group for our volunteers [54] and a list of de-identified comments received online between March 2022 and Nov 2023  (Appendix E, from Line 693). 

Regards

Dr Mark de Souza

Reviewer 3 Report (New Reviewer)

Comments and Suggestions for Authors

The paper is very well structured and its content is correctly organized. The main topic (addressed in the paper) although not novel, is relevant institutions, administration, staff, and patients. It is quite notable the way the paper goes through the different subjects related to the main question and it is easy to understand the way the selected methodology was applied. Results are also based on the developed research. As such, the only topic the paper could add more insights into would be how the configuration of the spatial system can contribute to improving the obtained results. This would imply an extra level of analysis that could bring more robustness to the major achievements of the paper. Nevertheless, this is not a field set to be explored in such a depth level, therefore, the paper is OK for publication as it is.

Author Response

Dear Reviewer

Thank you for your comments and suggestion regarding the configuration of the spatial system.

As a result we have now added Figure 8 to the manuscript which illustrates the network of walking paths and their relationship to the amenity being created by our planting zones.  The consumer request for this planting (as a cooling strategy) is referred to in Lines 357-359 in the section on the “Chalk the Campus” hot-spot mapping event. The location of these pathways maximises opportunities for campus users to have recurrent biophilic encounters as they traverse the campus on their usual journeys.

Kind regards
Dr Mark de Souza

This manuscript is a resubmission of an earlier submission. The following is a list of the peer review reports and author responses from that submission.

Round 1

Reviewer 1 Report

Comments and Suggestions for Authors

Thank you for the opportunity to review the paper “Healthy patients, workforce and environment (H3 Project): coupling climate action and mitigation to wellbeing in healthcare”.

Please see below my comments on this manuscript.

Introduction

Line 45: the authors mention the H3 project in this paragraph of Introduction. However, they did not detail the role/importance/reason of mentioning the project name here.

Chapter “Hospital operational pressures, built environment and local climate” is not clear. I did not understand what the authors aimed to emphasize in this section. Is the importance of H3 project? What is the relationship between H3 project, climate change and biophilic design? In addition, is there an influence of H3 project on Indigenous Territory (mentioned in the next section “Indigenous perspectives and context”)?

Overall, from this section, the reader could not clearly understand what are the aim and the objectives of the paper. Why is this paper unique/original?

Literature review is missing.

Line 113: is there a first hypothesis posed previously in the paper?

How did the authors measure their results? The method is not clearly explicated.

Author Response

Introduction

"H3 Project" was included to show the origins of the project as an innovative proposal to hospital health executives to encourage them to support climate adaptation and decarbonisation in the hospital (environmental benefit) that simultaneously promoted wellbeing of the health workforce and patients. The term also underpins the value of nature-based solutions and finding health cobenefits for climate action (including active mobility and promoting psychological wellbeing).

I believe the answers to the reviewer's questions (link between human health and biophilic design and benefits for Australian Indigenous hospital health outcomes) lie in the Discussion section.  There are multiple citations to validate these links. 

We have now added a table on local climate trends in our region which show a risk of heat stress for staff and patients.  The trends for warming in our region (due to climate change) are already included in the manuscript (Introduction, lines 31-44 ).

Climate-Health discourse is an emerging topic of interest.  This project is one of the first of its kind in Australia and is based on the growing evidence for the value of biophilic encounters in healthcare settings (these are amply referenced in the manuscript).

Measurement of Results: Details for surface temperature are covered in lines 136-143 and lines 239-230.  This information has been deliberately placed in Discussion to improve the understanding of this technical component for non-expert readers (clinicians and health policy writers).

The authors are transparent that the manuscript provides only early qualitative data through emails and social media posts.  Formal collection of qualitative data is proposed in Limitations and future directions (from line 582).  A followup manuscript is planned when the project has matured to allow more detailed study of planted and non-planted sites.

Literature review is covered in References Lines 639-856

Line 113 re "second hypothesis".  The authors included this term to indicate a sense of contemplation rather than pose a formal scientific hypothesis.  We are agreeable to changing the wording to: " In time we hope to explore whether the H3 Project informs and engages participants enough to encourage personal sustainability actions in their homes and social networks, the marketplace and their electorates".

Reviewer 2 Report

Comments and Suggestions for Authors

Thank you for the opportunity to review this paper. This topic is of high interest and very significant for the healthcare and Indigenous studies. However, I would recommend some improvements of the text:

(i) Abstract: please, specify the methods, sample and objectives.

(ii) Keywords: The text meeting two keywords ("Indigenous cultural safety" and "climate adaptation") are poor presented in the text. Please, remove them or develop these topics in the main text. 

(iii) Introduction. Climate change is a global issue. Therefore I would recommend including some references to international peer-reviewed journals and initiatives. The Introduction does not look as systematic introducing the main challenges. The objectives are missing.  

(iv) Material and methods: What is the period (month, year) of the study? Where (names of a place/places) did you conduct your study? Please, be more detailed in the description of data collection. Improve methodology of survey studies. If you conducted surveys, please, provide details about your survey process, sample size, inclusion criteria, questions (questionnaire as a supplementary is welcomed). Did you collect an informed consent from the participants?

(v) Results: The Results are not presented correctly. Actually, Results and Discussion are mixed. You should either clearly present the results and move some parts of the text to the Discussion section or make joint sections Results and Discussion.

(vi) Discussion: I would strongly recommend to move the text from the Results here. In the end of this section the authors should signify the strengths and limitations of their study.

Figure 5: use small letters a), b), c). Extend legenda of the table. 

Table 2 should be re-worked. It's too big and poor formated. 

Table 3 does not look like a table. Possibly, it can be presented as a text.

(vi) Discussion: What are the main findings of your study? Please, make the text of the Discussion more systematic.

Conclusion: almost missing. Please, respond to the objective (missing now) of your study.

The moderate English editing can improve the text.

Author Response

The authors respectfully point out the the manuscript documents a Case Study and is not a research project.  Therefore we believed that some of the reviewer's comments (such as the absence of method, subject and objectives) are not applicable.

Keywords: We acknowledge that inadequate information is included in the current manuscript.  In the interests of brevity, we will remove indigenous cultural safety and climate adaptation from the list of keywords

We concede that some content would be better placed in the discussion section instead of the results section.  We will action this in our revised manuscript.

We agree that climate change is a global issue however our study seeks to show application to the health care context. There are multiple references to the health benefits of biophilic spaces in health campuses, however our project is showing early evidence of an additive benefit of climate adaptation (22 degrees cooling).

We reiterate that our manuscript is a Case Study and have declared openly that only initial heat data and qualitative feedback is currently available.  As the plantings mature we intend to conduct more comprehensive heat collection and more formal qualitative research with appropriate surveys and ethics considerations.

We will modify the figure and tables as suggested in the next edition of the manuscript, including moving the large table (tree species) to an appendix.

We will take the reviewer's feedback on board and amend the conclusion and discussion section in our next version.

Reviewer 3 Report

Comments and Suggestions for Authors

·        Summary

The manuscript is a case report on a project on the articulation of healthy patients, workforce and environment, that was implemented at an Australian hospital.

·        Appraisal

I am of the opinion that the manuscript, being a case report, meets all the conditions for publication, in terms of its content and form. I almost regret the fact that I cannot recommend anything that could help improve the manuscript, as a case report. If it were an article, I would certainly like to see something related to the global burden of diseases associated with high temperatures. In any case, although the authors refer to some numbers at the beginning of the manuscript, I believe that a simple graph with the evolution, throughout the year, of temperatures and humidity in Darwin would help, especially non-Australians, to better understand the severity of the situation, which justifies resorting to projects such as the one reported.

Author Response

Thank you for your feedback and endorsement of publication 

We will include a table demonstrating the local annual heat and humidity profile for our region to better convey the issue to a global audience.